# Leveraging Large Language Models for Optimised Coordination in Textual Multi-Agent Reinforcement Learning

## Abstract

Cooperative multi-agent reinforcement learning (MARL) presents unique challenges, amongst which fostering general cooperative behaviour across various tasks is critical. Recently, large language models (LLMs) have excelled at dealing with challenges in the general RL paradigm, showcasing remarkable sample efficiency and adaptability across tasks through domain specific fine-tuning, or functional alignment. However, neither LLMs nor these fine-tuning approaches are designed with coordination-centric solutions in mind, and the challenge of how to achieve greater coordination, and hence performance, with LLMs in MARL has not yet been tackled. To address this, we introduce the 'Functionally-Aligned Multi-Agents' (FAMA) framework. FAMA harnesses LLMs' inherent knowledge for cooperative decision-making via two primary mechanisms. Firstly, it aligns the LLM with the necessary functional knowledge through a centralised on-policy MARL update rule. Secondly, it recognises the pivotal role of communication in coordination and exploits the linguistic strengths of LLMs for intuitive, natural language inter-agent message-passing. Evaluations of FAMA in two multi-agent textual environments, namely BabyAI-Text and an autonomous driving junction environment, over four coordination tasks show it consistently outperforms independent learning LLMs and traditional symbolic RL methods.

## 1 Introduction

A central paradigm of multi-agent systems (MAS) research is cooperative MAS, where agents are required to cooperate with each other to reach a desired outcome. The exact mechanism through which agents must coordinate is task-dependent, and can range from agents taking the same action in simple matrix games settings (Claus & Boutilier, 1998) to complex autonomous driving tasks (Slumbers et al., 2023) where agents must act in certain ways to not to crash into one another. In the multi-agent reinforcement learning (MARL) setting, there exists an extensive literature that proposes approaches to encourage coordination between agents (Oroojlooy & Hajinezhad, 2023). However, these generally suffer from a collection of problems: 1) learning is not sample-efficient or generalisable in the online-setting, 2) it is difficult to encourage general cooperative behaviour across a suite of tasks and 3) cooperative mechanisms are not (easily) human interpretable (Lazaridou et al., 2020).

It has been recently shown that using large language models (LLMs) as agents in single-agent reinforcement learning (RL) provides a potential solution to problem 1, and aspects of problem 2 (Carta et al., 2023). With respect to problem 1, LLMs encode sophisticated concepts as dictated by their understanding of language, therefore changing the need from learning fundamental concepts to aligning their prior knowledge to the functional requirements of the environment, allowing for sample-efficient learning (Carta et al., 2023). Furthermore, in relation to problem 2, as LLMs already encompass some form of fundamental knowledge, they are well suited to handling distributions of tasks that may only differ in slight ways (often where learning methods such as symbolic RL fail). However, these approaches are inherently single-agent, and have not been designed with coordination-centric solutions in mind, crucially not fully solving problem 2 or considering problem 3. The goal of this paper is to begin tackling both of these problems within the framework of LLMs as agents in MARL environments. In terms of problem 2, we focus on aligning LLMs to the functional

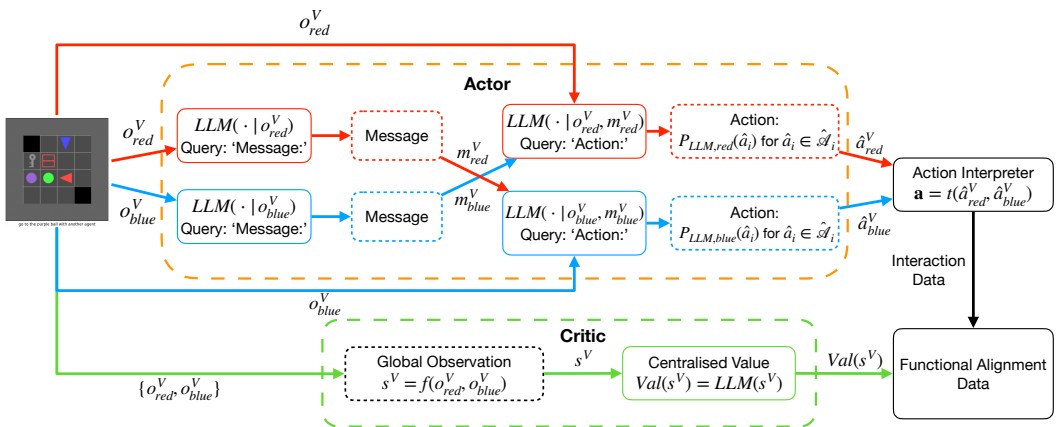

Figure 1: A visualisation of FAMA. Observations from the environment pass through both an Actor and a Critic, both providing the required data for the functional alignment process.

requirements of a multi-agent environment that requires coordination. In terms of problem 3, we leverage the notion that coordinated behaviour in humans is generally underpinned by some form of communication (Sally, 1995). MARL has explored using communication mechanisms between agents for cooperative tasks (Zhu et al., 2022; Karten et al., 2023; Singh et al., 2018), however they generally suffer from two issues: 1) explicit mechanisms need to be learned fresh for new environments and 2) these mechanisms are likely not to be easily human interpretable. We propose that an LLM agent can circumvent both of these issues by using natural language as an intuitive and interpretable communication mechanism.

To this end, this paper introduces the Functionally-Aligned Multi-Agents (FAMA) framework, the first generation of coordination-focused LLMs for MARL. Functional alignment references the idea that, whilst LLMs maintain diverse pre-trained knowledge, they are not necessarily able to directly convert this to the functional needs of an environment. We take an online fine-tuning approach that *aligns* the model's pre-trained knowledge with the *functional* requirements of the multi-agent environment that it interacts with. We allow one LLM to be the backbone of multiple agents, whilst centralising a Critic function in order to improve the coordination capabilities of the agents. This requires a series of changes including centralising the LLM architecture, the action-prompt design to distinguish between multiple agents and the fine-tuning update rule required. In addition, we allow the agents to communicate with each other in natural language, rather than other more abstract forms of communication like communication vectors (Sukhbaatar et al., 2016; Singh et al., 2018) or message tokens (Karten et al., 2023). This seemingly solves our two communication problems, natural language does not need to be learned fresh over new environments, and is completely human interpretable. FAMA therefore embeds a module designed strictly for communication between agents with the goal of improving coordination.

We answer the following two key questions when it comes to utilising LLMs in MARL:

1. What architectural and/or optimisation changes are required to foster a more coordination-centric solution for LLM MARL agents? How is overall performance impacted with respect to these changes, is coordination improved and error-making reduced?

2. Is natural language communication between agents a useful tool for improving coordination? Are the communication protocols that are arrived at by the models interpretable?

In order to answer these questions, we employ FAMA in two different textual MARL environments, our own multi-agent extension to BabyAI-Text (Carta et al., 2023; Hui et al., 2020) which generalises the BabyAI-Text single-agent tasks to multi-agent coordination variants, and a traffic junction environment (Sukhbaatar et al., 2016) where coordinated actions must be taken by the agents to avoid crashes. We answer Question 1 by evaluating FAMA over a series of tasks in the aforementioned environments against independent LLM approaches and symbolic baselines. To answer Question 2,

we specifically compare FAMA with and without a communication module, and focus on both the changes in performance and what communication actually occurs.

## 2 RELATED WORK

**Language-conditioned RL** This work falls generally into the field of language-conditioned MARL. At a high-level, language-conditioned RL involves agents that takes actions based on a language instruction / goal (Luketina et al., 2019). There exists a wide range of work that studies this language-conditioned RL setting in 2D or 3D environments (Colas et al., 2020; Chevalier-Boisvert et al., 2018; Küttler et al., 2020). In the MARL domain, language-conditioned algorithms is less well-understood, with Li et al. (2023); Ding et al. (2023) being two works that aim to solve some of the inherent problems of single-agent algorithms in multi-agent domains. Notably, these approaches do not directly use an LLM as the agent in the environment, but rather focus on taking language goals and grounding them to the environment, rather than interacting only in a text domain.

**Foundation Models and Decision Making** Self-supervised foundation models have proved to be incredibly powerful at knowledge transfer over a range of downstream tasks (Bommasani et al., 2021), and are increasingly being applied to more complex problems such as control (Brohan et al., 2022), planning (Huang et al., 2022b) and long-term reasoning (Wei et al., 2022). Notably, the intersection of foundation models in sequential decision-making problems is largely new. In the field of robotics, foundation models have been fairly extensively used as high-level planners (Huang et al., 2022b; Ahn et al., 2022; Liang et al., 2023). However, utilising an LLM as a high-level planner is fundamentally different from our goal, where the LLM directly takes actions in the environments. Another branch of works focus on fine-tuning either via behaviour cloning or offline RL (Wang et al., 2022; Reid et al., 2022; Takagi, 2022), which remains distinct from our work which only utilises online data. The closest work to ours is Carta et al. (2023), in which they also use online RL to ground an LLM as an action taker in text-worlds. The critical distinction is that we are operating in the multi-agent setting which requires fundamentally different considerations in the framework design.

## 3 PROBLEM FORMULATION

We consider a text-only augmentation of partially observable Markov games (POMG) (Liu et al., 2022). Therefore, we begin by defining a POMG, and then define the text-only elements that augment it. We denote an episodic POMG with $n$ agents by the tuple $(T, \mathcal{S}, \{\mathcal{A}_i\}_{i=1}^n, \{\mathcal{O}_i\}_{i=1}^n; \mathbb{P}, \mathbb{O}, \mu_1; \{r_i\}_{i=1}^n)$, where $T$ denotes the length of each episode, $\mathcal{S}$ the state space, $\mathcal{A}_i$ denotes the action space for the $i$-th agent. We denote by $\boldsymbol{a} := (a_1, ..., a_n)$ the joint action of all $n$ agents, and by $\mathcal{A} := \mathcal{A}_1 \times ... \times \mathcal{A}_n$ the joint action space. $\mathbb{P}$ is the transition matrix, so that $\mathbb{P}(\cdot|s, \boldsymbol{a}) \in \Delta_{\mathcal{S}}$ gives the distribution of the next state if joint action $\boldsymbol{a}$ is taken at state $s$. $\mu_1$ denotes the distribution of the initial state $s_1$. $\mathcal{O}_i$ denotes the observation space for the $i$-th agent. We denote by $\boldsymbol{o} := (o_1, ..., o_n)$ the joint observation of all $n$ agents, and the joint observation space by $\mathcal{O} := \mathcal{O}_1 \times ... \times \mathcal{O}_n$. $\mathbb{O}$ is the emission matrix, such that that $\mathbb{O}(\cdot|s) \in \Delta_{\mathcal{O}}$ gives the emission distribution over the joint observation space $\mathcal{O}$ at state $s$. Finally, $r_i$ is the collection of known reward functions for the $i$-th agent, so that $r_i(o_i)$ gives the deterministic reward received by the $i$-th agent if she observes $o_i$. In a POMG, at least some part of the state is always hidden from all agents, and each agent only observes their own individual observations and actions. At the beginning of each episode, the environment samples $s_1$ from $\mu_1$. At each step $t \in [T]$, each agent $i$ observes her own observation $o_{i,t}$, where $\boldsymbol{o}_t := (o_{1,t}, ..., o_{n,t}$ are jointly sampled from $\mathbb{O}(\cdot, s_t)$. Then each agent $i$ receives reward $r_i(o_{i,t})$ and picks action $a_{i,t} \in \mathcal{A}_i$ simultaneously. After that the environment transitions to the next state $s_{t+1} \sim \mathbb{P}(\cdot|s_t, \boldsymbol{a}_t)$ where $\boldsymbol{a}_t := (a_{1,t}, ..., a_{n,t})$.

We introduce multiple text objects to form a Text-POMG. Our goal is to allow LLMs to interactive with an underlying POMG. We denote a Text-POMG with $n$ agents by the tuple $(T, \mathcal{S}, \{\mathcal{A}_i\}_{i=1}^n, \{\mathcal{O}_i\}_{i=1}^n; \mathbb{T}, \mathbb{O}, \mu_1; \{r_i\}_{i=1}^n, \mathcal{V}, K, \{\mathcal{A}_i^{\mathcal{V}}\}_{i=1}^n, \{\mathcal{O}_i^{\mathcal{V}}\}_{i=1}^n, \mathbb{C}^{\mathcal{A}}, \mathbb{C}^{\mathcal{O}}, \{\mathcal{G}_i^{\mathcal{V}}\}_{i=1}^n)$. All tuple objects of the original POMG remain unchanged. $\mathcal{V}$ denotes the vocabulary space, with $v \in \mathcal{V}$ representing the available words. $K$ is the maximum text length, and can be thought of as the size of the context window of an LLM. $\mathcal{A}_i^{\mathcal{V}}$ is the text-action space of the $i$-th agent as described in terms of words $v \in \mathcal{V}$. We denote by $\boldsymbol{a}^{V} := (a_1^{V}, ..., a_n^{V})$ the joint text-action space of all $n$ agents, and by $\mathcal{A}^{\mathcal{V}} := \mathcal{A}_1^{\mathcal{V}} \times ... \times \mathcal{A}_1^{\mathcal{V}}$ the joint text-action. $\mathbb{C}^{\mathcal{A}}$ is a deterministic function that takes as

input joint text-actions $\boldsymbol{a}^V$ and converts them to environment joint actions $\boldsymbol{a} = \mathbb{C}^{\mathcal{A}}(\boldsymbol{a}^V)$. $\mathcal{O}_i^{\mathcal{V}}$ is the text-observation space of the $i$-th agent as similarly described in terms of words. We denote by $\boldsymbol{o}^V := (o_1^V, ..., o_n^V)$ the joint text-observation of all $n$ agents, and by $\mathcal{O}^{\mathcal{V}} := \mathcal{O}_1^{\mathcal{V}} \times ... \times \mathcal{O}_n^{\mathcal{V}}$ the joint text-observation space. $\mathbb{C}^{\mathcal{O}}$ is a deterministic functions that takes as input joint observations $\boldsymbol{o}$ and converts them to text-observations $\boldsymbol{o}^V = \mathbb{C}^{\mathcal{O}}(\boldsymbol{o})$. $\mathcal{G}_i^{\mathcal{V}}$ is the text-goal space for the $i$-th agent which is described in terms of words $v \in \mathcal{V}$. We denote by $\boldsymbol{g}^V := (g_1^V, ..., g_n^V)$ the joint text-goal of all $n$ agents, and by $\mathcal{G}^{\mathcal{V}} := \mathcal{G}_1^{\mathcal{V}} \times ... \times \mathcal{G}_n^{\mathcal{V}}$ the joint text-goal space. The reward functions are conditioned on a text-goal, such that agent $i$ observes their own text-observation $o_{i,t}^V$ and receives $r_i(o_{i,t}^V, g_i^V)$.

## 4 FAMA FRAMEWORK

FAMA is comprised of two core components, as demonstrated in Fig. 1, these are:

1. An Actor that provides a categorical distribution over discrete actions given an observation. In FAMA this will be dictated by the LLM and is discussed in Sec. 4.2.

2. A Critic that provides values given states or observations $V_\phi : \mathcal{S} \to \mathbb{R}$. In FAMA, a separate Critic head is used and is discussed in Sec. 4.4.

FAMA builds on top of these two components, giving it three key characteristics that we believe promotes solving problems 2 and 3, which are described further in this section. The first is the sharing of a singular LLM Actor between agents (Sec. 4.2), the second is the presence of a communication module (Sec. 4.3) and the third is a centralised Critic for training (Sec. 4.4).

### 4.1 PROMPT FUNCTION

The first requirement for any LLM-based agent framework is a prompt for agent $i$, $p_i^V$. In our setting, the prompt contains relevant information such that an LLM agent can take actions to maximise rewards. This is generally analogous to an observation in the RL setting, which contains the information an RL policy requires to take actions. We define an action prompt function $\rho_A : \mathcal{O}_i^{\mathcal{V}} \times \mathcal{G}_i^{\mathcal{V}} \times \mathcal{A}_i^{\mathcal{V}} \times \mathcal{C}_i^{\mathcal{V}} \to \mathcal{V}$ that takes as input an agent $i$'s text-observations, text-goals, text-actions and agent identifiers to construct a prompt for the LLM. For example, given the above information at time-step $t$, the prompt for the agent is defined as $p_{i,t}^V = \rho_A(o_{i,t}^V, g_{i,t}^V, \mathcal{A}_{i,t}^{\mathcal{V}}, c_i^V)$ where $c_i^V$ is the agent identifier (not dependent on $t$) and is described in Sec. 4.2, and note that the set of actions $\mathcal{A}_{i,t}^{\mathcal{V}}$ is passed in to represent all the actions that can possibly be taken given an observation at step $t$. The exact composition of $p_{i,t}^V$ is determined by the specifics of function $\rho_A$ which are different for each environment, and will be detailed for each environment in Appendix A. The following are two examples of an Actor prompt and then a Critic prompt:

> **Instruction**: You are an agent in a multi-agent reinforcement learning environment. *<Brief environment description>*. You are given a goal which requires coordinated behaviour with other agents. You can take the following actions: *<$\mathcal{A}_{i,t}^V$>*. You must pick the best action based on your observation to achieve the goal.
>
> **Goal**: *<$g_{i,t}^V$>*
>
> **Observation**: *<$o_{i,t}^V$>*
>
> **Action**: *<LLM begins response here>*

> **Instruction:** You are a critic value function in a multi-agent reinforcement learning environment. *<Brief environment description>*. You are given a goal which requires coordinated behaviour between agent *<$c_i^V$>* and agent *<$c_j^V$>*. Given an observation containing information from both of the agents, you will provide a numeric value of the observation.
>
> **Goals**: Agent *<$c_i^V : g_{i,t}^V$>*, agent *<$c_j^V : g_{j,t}^V$>*.
>
> **Observation**: Agent *<$c_i^V : o_{i,t}^V$>*, agent *<$c_j^V : o_{j,t}^V$>*

## 4.2 FAMA ACTOR

In FAMA, we avoid the practical (computational) issues of maintaining multiple large active LLMs for each independent agent Actor. Instead, we utilise one LLM to represent all of the agent's actor networks. The individual nature of the Actors are instead captured by agent identifiers $c_i^V$. These agent identifiers update a prompt with the required agent specific information that allows the LLM to differentiate between the Actor that it is meant to be representing. Examples of $c_i^V$ could be: 'You are acting as Agent $i$', or 'Identity: Agent $i$, Role: Chef' dependent on the environment context. For our experiments, we provide the agent identifiers used in Appendix A.

To understand the workings of the Actor for agent $i$, we first outline the generation of discrete actions given a prompt as described in Sec. 4.1. At time-step $t$, agent $i$ receives a text-observation $o_{i,t}^V$ and a text-goal $g_{i,t}^V$. This is subsequently transformed into the text-prompt $p_{i,t}^V = \rho_A(o_{i,t}^V, g_{i,t}^V, \mathcal{A}_{i,t}^V, c_i^V)$ for agent $i$, where $\rho_A$ is defined in Sec. 4.1. We want the agent to take an action $a_{i,t}^V(p_{i,t}^V)$ that is some function of the prompt provided to it. There are multiple ways to generate actions based on a prompt $p_{i,t}^V$, for example in the single-agent setting Huang et al. (2022a); Li et al. (2022); Carta et al. (2023) propose methods such as free-text generation: given a prompt $p_t^V$, query the LLM to generate a sequence of output words $\hat{a}_t(p_t^V) = \{v_0, v_1, ..., v_K\}$, with $K$ being the max number of words, and each $v_k \in \mathcal{V}$. Then heuristically (e.g. parse the sequence for mentions of an action) interpret this sequence to select an action $a_t^V \in \mathcal{A}^{\mathcal{V}}$ such that for some $k$, $v_k$ corresponds to the chosen action $a_t^V$. Ahn et al. (2022) propose utilising $|\mathcal{A}^{\mathcal{V}}|$ action heads to compute the probability of each action $a_t^V \in \mathcal{A}^{\mathcal{V}}$ by computing for each $a_t^V$ the conditional probability of each of its constituent tokens $w_k$. However, Carta et al. (2023) find that creating individual action heads is not necessary, and it is more effective to just simply query the full LLM as the Actor. This is the approach that we adopt in order to generate actions in FAMA. Formally, the probability of agent $i$ taking the action $a_{i,t}^V$ is:

$$\mathbb{P}_{LLM}(a_{i,t}^V | \rho_A(o_{i,t}^V, g_{i,t}^V, \mathcal{A}_{i,t}^{\mathcal{V}}, c_i^V), \theta) = \prod_{k=0}^{|a_{i,t}^V|} \mathbb{P}_{LLM}(w_{i,k} | \rho_A(o_{i,t}^V, g_{i,t}^V, \mathcal{A}_{i,t}^{\mathcal{V}}, c_i^V), w_{i,<k}, \theta) \quad (1)$$

where $|a_{i,t}^V| \in \mathbb{Z}^+$ is the $k$ number of tokens $w_k$ that make up the word $a_{i,t}^V$. The prompt for the LLM is $\rho_A(o_{i,t}^V, g_{i,t}^V, \mathcal{A}_{i,t}^{\mathcal{V}}, c_i^V)$ and is based on the text-observation, text-goal, available actions and agent identifier, and $\mathbb{P}_{LLM}(w_{i,k} | \rho_A(o_{i,t}^V, g_{i,t}^V, \mathcal{A}_{i,t}^{\mathcal{V}}, c_i^V), w_{i,<k}, \theta)$ represents the probability that token $w_{i,k}$ of action $a_{i,t}^V$ is selected given the prompt and the already selected tokens $w_{i,<k}$, and $\theta$ are the parameters of the Actor.

## 4.3 COMMUNICATION MODULE

A characteristic of human coordination is that we use words as a method to coordinate with each other to reach our goals (Wittgenstein, 1953; Austin, 1975; Clark, 1996). Therefore, we believe a natural step to improve coordination in our framework is to leverage our agents being language models, by introducing a form of natural language communication between agents. We envisage two useful reasons for a communication module: 1) Agent's in the system can share internal information about their own current goals / intent which can guide the actions of others and 2) natural language communication is human-interpretable and could act as a natural method to improve human-AI coordination in the future.

Agents generate a discrete message that is broadcast to other agents. At the start of a time-step $t$ each agent receives their own observation $o_{i,t}^V$, we allow the agents to communicate information based on $o_{i,t}^V$. The information communicated is intended to improve the ability of the agents to coordinate with each other. Inspired by Karten et al. (2023), we let the agent's communicate their *intent* based on $o_{i,t}^V$. This framework places no restriction on what message is communicated, for example allowing for the agents to generate free-text with a token limit of $K^m$ (generally $K^m << K$). However, we focus on discrete message selection in a similar manner to action selection. For agent $i$, we have a discrete set of potential text-messages $\mathcal{M}_{i,t}^V$. We define a message prompt function $\rho_M : \mathcal{O}_i^{\mathcal{V}} \times \mathcal{G}_i^{\mathcal{V}} \times \mathcal{M}_i^{\mathcal{V}} \times \mathcal{C}_i^{\mathcal{V}} \to \mathcal{V}$ that is mostly similar to $\rho_A$, but instead takes the set of possible messages rather than possible actions. The conditional probability of selecting each message $m_{i,t}^V \in \mathcal{M}_{i,t}^V$ is calculated by calculating the

constituent tokens being selected by the LLM given the prompt:

$$\mathbb{P}_{LLM}(m_{i,t}^V | \rho_M(o_{i,t}^V, g_{i,t}^V, \mathcal{M}_{i,t}^{\mathcal{V}}, c_i^V), \theta) = \prod_{k=0}^{|m_{i,t}^V|} \mathbb{P}_{LLM}(w_{i,k} | \rho_M(o_{i,t}^V, g_{i,t}^V, \mathcal{M}_{i,t}^{\mathcal{V}}, c_i^V), w_{i,<k}, \theta) \tag{2}$$

where $|m_{i,t}^V| \in \mathbb{Z}^+$ is the $k$ number of tokens $w_k$ that make up the word $m_{i,t}^V$. We then pass these messages between agents, such that the message of agent $i$ is passed to agent $j$ and vice versa. The actions that the agents then take are conditioned on not only the local observation $o_{i,t}^V$, but also the intentions of both of the agents. We define a new observation, $\hat{o}_{i,t}^V(o_{i,t}^V, m_{-i,t}^V)$, that wraps the messages from the other $-i$ agents into the environment observation. This allows us to calculate the probabilities of each discrete action for agent $i$, $P_{\text{LLM}}(a_{i,t}^V | m_{-i,t}^V) = \prod_{k=0}^{|a_i^V|} \mathbb{P}_{LLM}(w_{i,k} | \rho_A(\hat{o}_{i,t}^V(o_{i,t}^V, m_{i,t}^V), g_{i,t}^V, \mathcal{A}_{i,t}^{\mathcal{V}}, c_i^V), w_{i,<k}, \theta)$ for all $\hat{a}_{i,t}^V \in \mathcal{A}_{i,t}^{\mathcal{V}}$. We hypothesise that communicating intentions between agents will aid in improving coordination ability in environments where coordination is time-sensitive between agents, or there requires some separation of tasks between agents that needs to be decided.

## 4.4 FUNCTIONAL ALIGNMENT

Functional alignment is the process of taking an LLM's pre-trained knowledge and aligning it with the functional requirements of an environment. The alignment process, in our case, is an online RL approach that tunes the LLM parameters based on rewards from the environment. In order to functionally align our LLM agents, we propose using the MAPPO Yu et al. (2022) update rule based on the on-policy data received from environment interaction. To do this, we require a Critic to provide a value given the environment state. For the Critic network $V_\phi$ we append the LLM with a 'Critic Head' on the last layer of the first Decoder block. The Critic Head is an MLP with a single numeric output designed to measure the value of the observation provided to it in text form. The goal of a Critic function is to evaluate states in order to improve the training process for the Actor, which can then be employed in a decentralised fashion. Therefore, in order to improve the coordinated nature of our Actors we utilise a centralised Critic that takes information from all of the agents to generate a global observation. The Critic then approximates a value function over the global observation, valuing states where the agents are closer to achieving their coordinated goal higher than those not.

Finally, the Actor is trained to maximise the following objective:

$$L(\theta) = \left[ \frac{1}{Bn} \sum_{k=1}^{B} \sum_{i=1}^{n} \min\left( \eta_{\theta,i}^{(k)} A_i^{(k)}, \ \text{clip}\left( \eta_{\theta,i}^{(k)}, 1 - \epsilon, 1 + \epsilon \right) A_i^{(k)} \right) \right]$$
$$+ \left[ \sigma \frac{1}{Bn} \sum_{k=1}^{B} \sum_{i=1}^{n} S\left[ \mathbb{P}_{LLM}\left( \hat{p}_i^V\left( p_i^{V,k}, g_i^V, c_i^V, m_{-i}^V, \theta \right) \right) \right] \right]$$

where $\eta_{\theta,i}^{(k)} = \frac{\mathbb{P}_{LLM}(a_{i,k}^V | p_{i,k}^V, m_{-i,k}^V, \theta)}{\mathbb{P}_{LLM}(a_i^{(V,k)} | p_i^{V,k}, g_i^V, c_i^V, m_{-i}^V, \theta_{old})}$, $B$ is the batch size, $A_i^{(k)}$ is the advantage computed via GAE, $S$ is the policy entropy and $\sigma$ is the entropy coefficient hyperparameter. In a similar manner we can optimise the Critic head by minimising the following loss function:

$$L(\phi) = \frac{1}{Bn} \sum_{k=1}^{B} \sum_{i=1}^{n} \max\left[ \left( V_\phi(s^V) - \hat{R}_k \right)^2, \right.$$
$$\left. \left( \text{clip}\left( V_\phi(s^V), V_{\phi_{old}}(s^V) - \epsilon, V_{\phi_{old}}(s^V) + \epsilon \right) - \hat{R}_k \right)^2 \right]$$

where $V_{\phi_{old}}$ is a target Critic head and $\hat{R}_k$ is the discounted reward-to-go. Due to the nature of the critic function being shared, there is no need for agent identifier conditioning in the value function as all local observations $o_i^V$ constitute the global observation $s^V$.

## 5 EXPERIMENTS

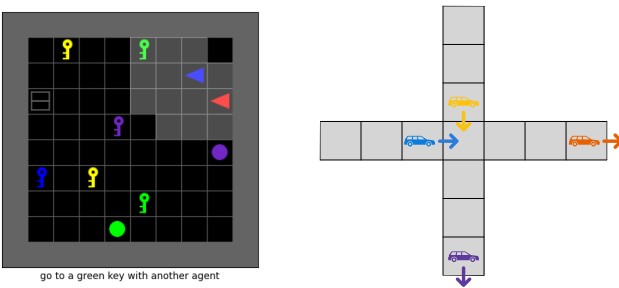

Figure 2: a) An instance of the multi-agent BabyAI-Text environment on the Go-To task. b) Visual representation of the junction environment, where arrows display the direction of car movement.

We presented two main questions that we would like to address: 1) What changes are required to improve coordination in the framework of LLM agents in MARL, and the empirical consequences of these changes and 2) how natural language communication impacts performance, alongside its interpretability. In this section, we will empirically evaluate FAMA under the scope of these two questions. First, we introduce our baselines, then we overview the environments and tasks that we study. After, we present our results with respect to the two questions. Over these experiments, we use a mix of Flan-T5 (Rae et al., 2021) model sizes as our LLM (exact size selection for each experiment in Appendix B.1).

### 5.1 BASELINES

As noted previously, we believe our work is the first to strictly utilise an LLM as an agent in the strictly multi-agent setting. Therefore, whilst there does not exist an exact baseline, we propose to utilise the following three algorithms similarly to Carta et al. (2023):

**Independent GLAM** Carta et al. (2023) - GLAM is an LLM framework that also directly acts as a agent policy, but is designed strictly for single-agent settings. We allow each of the agents to act independently using GLAM updates.

**Symbolic IPPO** - An independent PPO algorithm that instead runs on the symbolic version of the BabyAI environment Hui et al. (2020).

**Symbolic MAPPO** - A multi-agent PPO algorithm that instead runs on the symbolic version of the BabyAI environment Hui et al. (2020).

### 5.2 MULTI-AGENT BABYAI-TEXT ENVIRONMENT

We introduce a general set of multi-agent coordination tasks built on top of the BabyAI-text Carta et al. (2023) environment. Specifically, we propose three multi-agent coordination tasks:

**Go to ⟨object⟩ with another agent** - A simple navigation task that requires reasoning abilities to choose the right route given objects' position. In addition, agents will only receive reward if they both go to the object at the same time. An example of this can be seen in Fig. 2a.

**Go to ⟨object⟩ with another agent (Punishment)** - A simple navigation task that requires reasoning abilities to choose the right route given objects' position. In addition, agents will only receive reward if they both go to the object at the same time. Agents will be punished with negative reward if they go to the object without the other agent also being there.

**Pick up ⟨object⟩** - A reasoning task combined with a navigation task. Agents receive reward if they perform the pick-up action when facing towards the object. In the multi-agent variant, both agents must perform the pick-up action facing the object at the same time to receive reward.

Whilst these tasks are simple, they are sufficient to demonstrate performance differences between our method and the baselines. In addition, the tasks that are solved by these algorithms will improve drastically as larger LLMs become more computationally feasible to run and functionally align.

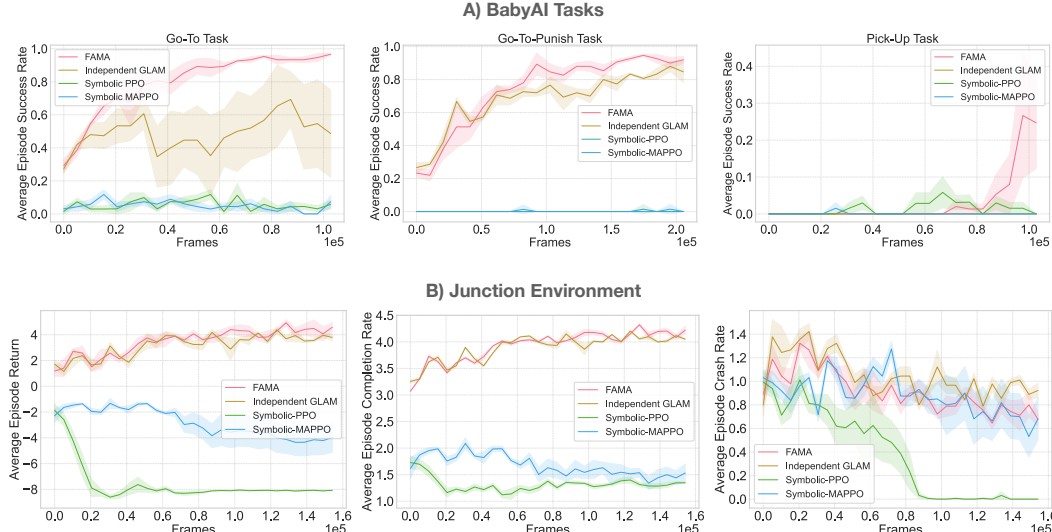

Figure 3: Results for tasks in the multi-agent BabyAI-Text environment and traffic junction environment. The first row represents average episodic results (50 Episodes) over 5 training seeds for each of the algorithms. The second row represents the same for the traffic junction environment. Higher metrics are better, other than for Average Episode Crash Rate in B).

## 5.3    TRAFFIC JUNCTION ENVIRONMENT

Introduced in Sukhbaatar et al. (2016), the traffic junction environment consists of a $g \times g$ grid. At each time step, new cars enter the grid with probability $p_{arrive}$ from each of the available $d$ directions. The goal is to reach the end of the road without crashing into the other cars. Cars are also encouraged to reach their goal as quickly as possible, as they receive negative rewards per time-step on the grid. Each car occupies a single cell at any given time, and they have the options of either moving forward or staying still at each time-step. Full environment details are provided in Appendix A.

## 5.4    RESULTS

### 5.4.1    Q1. DOES FAMA FOSTER A COORDINATION-CENTRIC SOLUTION?

In Fig. 3 we demonstrate the performance metrics of FAMA (note in this case we are not using the communication module) versus our baselines. Fig. 3A shows the results on the multi-agent BabyAI-Text environment. In all of the tasks, FAMA achieves both the best performance in terms of rewards (some tasks have bonuses / penalties) and overall success rate. In the hardest task, the multi-agent Pick-Up Task, independent GLAM is generally entirely unable to complete the task whereas FAMA begins to learn towards the end of training to successfully complete the task in around 30% of episodes. Notably, the symbolic baselines generally fail without learning anything. The most likely reason for this is due to the poor sample efficiency of the algorithms, with the amount of steps needed for the LLM to learn useful behaviour being far too few for the symbolic approaches.

In terms of the traffic junction environment, we have a similar story, FAMA outperforms all of the baselines, however to less of an extent against independent GLAM as in the multi-agent BabyAI-Text tasks. The major noticeable difference in the performance of the two algorithms is in the crash rate, where FAMA displays an average crash rate per episode decrease of 24.7% over independent GLAM. This increased level of error avoidance is a particularly nice property in a coordination system.

## 5.5    Q2. IS NATURAL LANGUAGE COMMUNICATION BENEFICIAL?

During our experimental process, we found that the communication module was particularly useful in the presence of penalties. For example, in our Go-To-Punishment task where agents are punished if they arrive at the goal without the other agent, or in our traffic junction environment where agents are punished for crashing. In Fig. 4 we demonstrate these results. In Fig. 4a we show the error rates of FAMA and FAMA-Communication alongside their corresponding success rates on the Go-To-

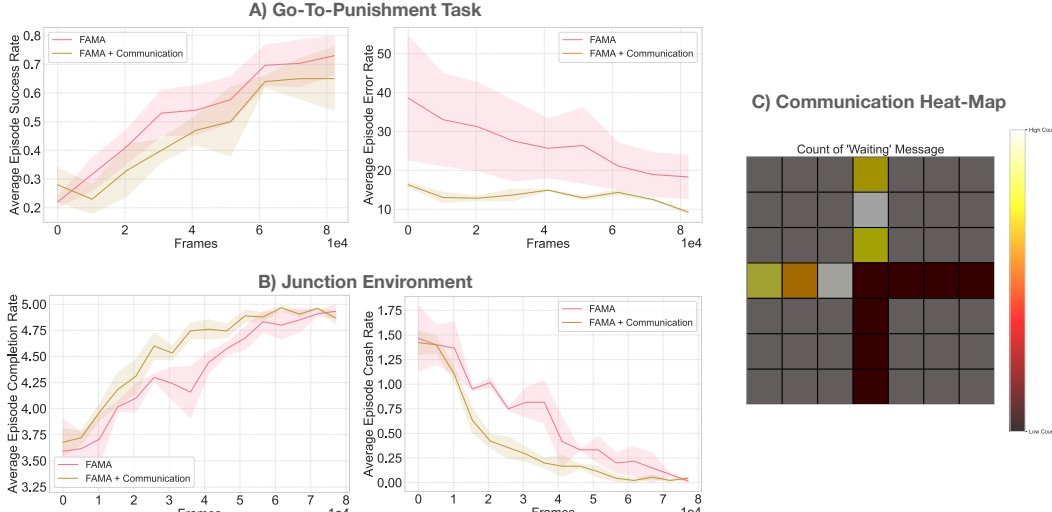

Figure 4: Results of FAMA with and without the communication module. A) Represents average episodic return (50 Episodes) over 5 training seeds on the Go-To-Punishment task. B) Same but on the traffic junction environment. C) A heat-map for the traffic junction environment demonstrating when the 'waiting' message is communicated, over 100 episodes. Higher metrics are better for Success/Complete Rate, lower metrics are better for Error/Crash Rate.

Punishment task. We first notice that, in terms of final success rates both variants arrive at similar values, with slight under-performance from FAMA-Communication. However, from the beginning of the training process the amount of errors that FAMA-Communication makes is much lower on this task, and remains that way for the whole of the training process.

Similarly, in Fig. 4B, we show a similar analysis in terms of the crash rate in the traffic junction environment. We note that the analysis is slightly different in this case, as FAMA without a communication module does arrive at a final crash rate of similar to FAMA-Communication. However, FAMA-Communication is far more sample-efficient in terms of reducing the crash rate and very early into training is able to significantly reduce its crash rate. In Fig. 4c we provide a heat-map demonstrating an initial investigation into what the LLM has learned in terms of a messaging system. Note that the agents arrive at the top and the left of the grid, and once they are passed the middle square in the grid messages become meaningless and are subsequently ignored. The heat-map shows at which grid-step the agents sent the 'waiting' message to the other agent. The light grey spots represent the highest density of messaging, showing that the agents learn to predominately message in a different grid-spot from each other. This firstly makes sense, as for example, both players messaging 'waiting' at the entrance to the junction will cause disputes, therefore leading to one agent taking control of messaging in this grid-spot. This leads to the other agent providing its 'wait' message one grid-spot earlier to 1) minimise confusion upon arriving at the junction and 2) informing the other agent that they are close to the junction.

## 6    CONCLUSION

We introduce FAMA, a framework for utilising and training coordination-centric LLMs in MARL environments. FAMA relies on three core components to foster coordination, the dual centralisation of both the agents within one LLM and the centralised Critic function, and a natural language communication module allowing agents to interact with each other. The dual centralisation within FAMA proves powerful in comparison to independent or symbolic methods in promoting coordinated behaviour within agents in both multi-agent BabyAI-Text tasks and an autonomous driving traffic junction environment. Furthermore, the communication module demonstrates particular benefits in tasks that require accurate coordination timing, as it allows the agents to coordinate timing through natural language. Future work should focus on two main directions: 1) how to improve the inference time required to output actions for multiple agents as LLM sizes grow and 2) non-discrete communication allowing for a diverse range of messaging opportunities.

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

## A ENVIRONMENT DETAILS

### A.1 MULTI-AGENT BABYAI-TEXT

We introduce a general set of multi-agent coordination tasks built on top of the BabyAI-text Carta et al. (2023) environment. First, Chevalier-Boisvert et al. (2018) introduced BabyAI, a language-driven environment in which an agent has a finite amount of steps to fulfill a language objective. This platform utilizes a gridworld scenario (MiniGrid) to create a series of environments for following instructions. It's specifically tailored for studies on grounded linguistic learning and associated challenges with sample efficiency. The gridworld is filled with the agent and objects in 6 different colors: boxes, balls, doors, and keys. These items are situated in 8 × 8 tile rooms connected by doors, which can be either locked or shut. The grid's layout is determined procedurally, meaning objects and their positions, along with the agent's position, are chosen at random for each episode. Some objects are pertinent to the task, whereas others serve as distractions (the agent needs to navigate around or move them). The agent can execute 6 basic actions: turn left, turn right, move ahead, toggle, and pick up, which aid in completing the language directive, like "Pick up the red box". The agent's perspective is limited; it only perceives objects in the 6 × 6 grid directly in front of it. BabyAI offers this limited viewpoint via a symbolic representation composed of three 6 × 6 matrices. The first matrix identifies objects in the visible cells, the second notes their colour, and the third defines their status (e.g., locked, unlocked). Upon task completion in $N$ steps, the agent gets a reward $r_N = 1 - 0.9\frac{N}{H}$, with $H$ representing the maximum number of steps. During training, all rewards are increased 20-fold, as suggested by Mirchandani et al. (2021), to ensure efficient reward propagation. The agent receives a reward of 0 if the task remains incomplete in the given steps.

BabyAI-Text Carta et al. (2023) is a text-based setting which wraps around BabyAI and offers an observational description rather than a symbolic view. This description follows specific template guidelines:

- "You see a *<object> <location>*" if the object is a key, a ball, a box or a wall.
- "You see a(n) *open/closer* door *<location>*", if the agent sees a door.
- "You carry a *<object>*", if the agent carries an object.

Here, *<object>* combines an adjective (chosen from 6 potential colours: red, blue, green, yellow, grey, and purple) and a noun (from 4 possibilities: key, door, box, ball). The <location> denotes the number of moves right, left, or forward the agent would need to reach the object. For instance, as shown in the far-left observation of Figure 5, the "yellow box" is described as being "2 steps to the left and 1 step ahead" from the agent (represented by the red triangle). Therefore, an object depicted as "1 step ahead" is directly in the agent's path, meaning it doesn't need to move to retrieve it.

Expanding on BabyAI-Text, we introduce a series of multi-agent tasks. The key addition necessary is for a new observational description to be added:

- "You see a *<colour>* agent *<location>*", if an agent sees another agent.

### A.1.1 TASK DETAILS

Here we detail the new multi-agent tasks that we add to BabyAI-Text.

**Go to ⟨object⟩ with another agent** - A simple navigation task that requires reasoning abilities to choose the right route given objects' position. In addition, agents will only receive successful task completion reward if they both go to the object at the same time. An example of this can be seen in Fig. 2a.

**Go to ⟨object⟩ with another agent (Punishment)** - A simple navigation task that requires reasoning abilities to choose the right route given objects' position. In addition, agents will only receive reward if they both go to the object at the same time. Agents will be punished with negative reward if they go to the object without the other agent also being there, receiving $r_N == 0.05$.

**Pick up ⟨object⟩** - A reasoning task combined with a navigation task. Agents receive reward if they perform the pick-up action when facing towards the object. In the multi-agent variant, both agents

must perform the pick-up action facing the object at the same time to receive successful completion reward.

### A.1.2 PROMPT DETAILS

**Instruction**: You are an agent in a multi-agent reinforcement learning grid world. You are given a goal which requires coordinated behaviour with the other agent. You can take the following actions: go_forward, turn_left, turn_right, pick_up. You must pick the best action based on your observation to achieve the goal.

**Goal**: Go to the *<colour><object>* at the same time as another agent.

**Observation**: $<o_i^V>$

**Action**: *<LLM begins response here>*

### A.2 JUNCTION ENVIRONMENT

### A.2.1 TASK DETAILS

The traffic junction environment consists of a two-way junction on a 7x7 grid. At each time-step, new cars enter the grid with probability 0.75 from one of the two directions. The total number of cars at any given time is limited to 5. Each car occupies a single cell at any given time and is randomly assigned to one of the two possible routes. At every time step, a car has two possible actions: accelerate which advances it by one grid spot on its route, or brake which keeps it in its current grid spot. Cars are removed from the grid once it reaches the destination at the edge of the grid, providing a reward of $r = 1$. Two cars collide if they location overlaps, incurring a reward of $r = -10$, otherwise the simulation is unaffected and the cars continue on in grid, notably not receiving a reward for reaching the end of the road. To discourage slow driving, each car receives a reward of $r = -0.01\tau$ at every time-step $\tau$ since the car arrived on the grid. Each car is only able to observe its direct neighbourhood of grid spots, i.e. a $3x3$ grid around them.

Therefore, junction environment text provides text observations as:

- "You see the road one step ahead of you.", if the grid-spot in front of the agent is road.
- "You see the end of the road one step ahead of you.", if the grid-spot in front of the agent is the edge of the grid.
- "You see a car one step ahead and one step to the right of you", if the agent is at the junction entrance whilst another agent is at a different junction entrance.

### A.2.2 PROMPT DETAILS

**Instruction**: Instruction: You are an agent in a multi-agent reinforcement learning driving environment. Each agent is on a different road, and all the roads meet in the middle of the environment. You are given a goal which requires coordinated behaviour with other agents. You can take the following actions: go_forward which moves you forward one spot on the road, or stay_still which keeps you in the same spot on the road. You must pick the best action based on your observation to achieve the goal.

**Goal**: Get to the end of the road without crashing into another agent.

**Observation**: $<o_i^V>$

**Action**: *<LLM begins response here>*

## B MODEL DETAILS

### B.1 LLM

We utilise two different Flan-T5 (Rae et al., 2021) models in our experiments dependent on the environment.

For the multi-agent BabyAI-Text experiments we use functionally align Flan-T5 Base (248M Parameters). For the junction environment experiments not using communication we functionally align Flan-T5 Small (80M Parameters), and for the junction environment experiments with communication we funtionally align Flan-T5 Base (248M Parameters). We used 4x Nvidia V100 GPUs for all experiments, with Flan-T5 Base being split over two GPUs and Flan-T5 Small utilising one GPU.

## B.2 HYPERPARAMETERS

We do not perform any hyperparameter tuning, opting to use reuse the hyperparameters from (Ramamurthy et al., 2022). For the Critic heads we use MLPs with 3 hidden layers of 1024 units with Sigmoid activation.

Table 1: PPO hyperparameters

| Variable | Value |
|---|---|
| Number of transitions between updates | 1280 |
| Number of epochs per update | 4 |
| Batch size | 64 |
| Entropy loss coefficient | 0.01 |
| Value function loss coefficient | 0.5 |
| Discount factor | 0.99 |
| Learning rate | $1 \times 10^{-6}$ |
| $\lambda$ factor of GAE | 0.99 |
| Clipping parameter $\epsilon$ | 0.2 |
| Maximum gradient norm | 0.5 |

Table 2: Adam hyperparameters

| Variable | Value |
|---|---|
| Learning rate | $1 \times 10^{-6}$ |
| $\beta_1$ | $1 \times 10^{-5}$ |
| $\beta_2$ | 0.9 |
| Clipping parameter $\epsilon$ | 0.999 |

## B.3 BASELINE DETAILS

**Independent GLAM** - We directly use the implementation of GLAM (Carta et al., 2023) provided by the authors at `https://github.com/flowersteam/Grounding_LLMs_with_online_RL/blob/main/experiments/agents/ppo/llm_ppo_agent.py` for each of the independent agents. We do not alter any hyperparameters.

**Symbolic PPO** - We directly use the implementation provided at `https://github.com/flowersteam/Grounding_LLMs_with_online_RL/blob/main/experiments/agents/ppo/symbolic_ppo_agent.py` for each of the independent agents. Hyperparameters are provided in Table 1.

**Symbolic MAPPO** - We implement MAPPO (Yu et al., 2022) on top of the PPO implementation provided at `https://github.com/flowersteam/Grounding_LLMs_with_online_RL/blob/main/experiments/agents/ppo/symbolic_ppo_agent.py`. We implement a centralised critic that takes as input the partial observations of each agent. Hyperparameters are those listed in Table 1.

