# OpenReview forum: "Leveraging Large Language Models for Optimised Coordination in Textual Multi-Agent Reinforcement Learning"
_ICLR.cc/2024/Conference — Submitted to ICLR 2024_

### Official Review · Reviewer_dP6r · 2023-10-25

**Soundness:** 2 fair
**Presentation:** 2 fair
**Contribution:** 2 fair
**Rating:** 3
**Confidence:** 4

**Summary:**

The paper introduces the Functionally-Aligned Multi-Agents (FAMA) framework to improve coordination in Multi-Agent Reinforcement Learning (MARL) using Large Language Models (LLMs). FAMA innovatively employs LLMs for action selection and inter-agent communication, extends traditional game models to better suit text-based environments, and addresses key research questions about the role of LLMs and natural language in MARL. The framework aims to offer a structured approach for enhanced decision-making and coordination among agents.

**Strengths:**

1. The use of Large Language Models (LLMs) as policy mechanisms in Multi-Agent Systems (MAS) is highly innovative. This unique approach sets the paper apart and introduces a new dimension to the field.

2. The paper excels in the design aspects, particularly in the formulation of prompts. The detailed approach in this area adds depth and rigor to the research, enhancing its overall quality.

**Weaknesses:**

1. While the paper offers a novel approach by combining reinforcement learning and LLMs, the alignment strategy doesn't seem to differ significantly from past CTDE methods. This raises questions about the contribution of the work.

2. The paper employs a text-based environment, which limits the applicability of using LLMs as policies in general RL tasks where text cannot be directly used for actions with the environment. This constraint could limit the generalizability of the method.

3. While the paper shows that communication improves performance, it doesn't provide a comparative analysis to quantify how much better the performance is when using natural language for communication as opposed to non-natural language methods.

**Questions:**

1. If the method to solve the multi-agent problem still relies on the CTDE's credit assignment approach, then where does the advantage of LLMs manifest, apart from the part where it can communicate using natural language?

---

> ### Author Response · Authors · 2023-11-17
>
> ### We thank Reviewer cs97 for their efforts in offering constructive comments and questions.
>
> 1. > **Reviewer**: Contribution of the work.
>
> **Response**:
> Our approach is a CTDE method, and the major contribution of the work is formalising the necessary details of a CTDE framework for LLM RL agents. The main interest of the paper is to design a CTDE framework for LLM RL agents and we have shown an initial attempt at doing this, which we believe can be extended on easily by future work. In addition, the second major contribution of this work is introducing natural language communication into the CTDE framework, which is not possible in the traditional MARL setting.
>
> 2. > **Reviewer**: Generalizability of the method.
>
> **Response**:
> Whilst we limit this paper to only text-based environments, and would expect in its current iteration to only see obvious benefits in text-based environments over traditional RL approaches, we do not believe it is limited. In our opinion, there is essentially no limitation on the usage of LLM policies in general RL, simply because translating general RL tasks into text tasks is not strictly difficult. For example, in all general RL tasks that we are aware of, discrete actions have an obvious text counterpart (e.g. movement actions) that can be passed to an LLM. This just requires translation work from the practicioner.
>
> We envision this to be more problematic for the environment observation, which requires far more details in translating to text-only. For example, both of our environments are directly translated for general RL counterparts, however being grid environments makes the translation easier. A continuous environment will be more involved. An interesting research direction would be using multi-modal models to 'caption' frames in the environment to directly translate to observations, but this is outside the scope of this work.
>
> 3. > **Reviewer**: Non-natural language communication baseline.
>
> **Response**:
> We thank the reviewer for noting that communication does indeed improve performance. For the camera-ready version of the paper we will add the IC3Net baseline (Learning When To Communicate at Scale in Multiagent Cooperative and Competitive Tasks - Singh et al. ICLR 2019) which also experimented on the traffic junction problem. Our expectation is that, whilst their paper suggests strong performance (pushing towards the performance of ours), the details of their experimental setup are generally easier. For example, they use curriculum learning (we do not), train for longer, and the probability of new cars spawning into the environment is lower. Therefore, we expect to still maintain a gap in performance.
>
> We hope we have managed to address the concerns of the reviewer. If so, we ask that the reviewer would consider improving their score, or to point out other additional issues that we can attempt to clarify.

---

> > ### Comment · Reviewer_dP6r · 2023-11-23
> >
> > Thank you for your responses. Despite this, I still feel that the overall contribution has its limitations. Therefore, I have decided to maintain my current score.

---

### Official Review · Reviewer_cs97 · 2023-11-01

**Soundness:** 3 good
**Presentation:** 2 fair
**Contribution:** 3 good
**Rating:** 6
**Confidence:** 4

**Summary:**

The paper introduces a method(FAMA) facilitating coordination for textual multi-agent reinforcement learning by leveraging LLM. FAMA consists of an actor where LLM can be used to infer probability of each action, a communication module to enhance agent-to-agent coordination and a functional alignment step to fine-tune the LLM with a critic head.

The experiments results are promising and FAMA beats benchmarks  in most environments. Communication module is particularly studied where its sample efficiency is demonstrated and ablation study reinforces its contribution.

**Strengths:**

Applying LLM to RL in general is a relatively new and interesting topic. This work extends LLM to multi-agent setting where natural language exhibits a. nature role in "communication" among agents.

The experiments are not complicated environments but solid enough to me to demonstrate proposed method's superiority.

**Weaknesses:**

1. The paper is in general not well written, especially in section 4. Notation is difficult to understand and sometimes with ambiguity. For example, in section 4.2, what are parameters of \hat{p}_i_V? And in section 4.3, are those m_i^V are automatically generated or pre-selected and using LLM to get its likelihood? It might be better to give a couple of examples  for 4.2, 4.3 to demonstrate how those steps are conducted.

2. Since it's a paper without any theoretical justification, more environments results might be more convincing.

**Questions:**

Stated in weakness part.

---

> ### Author Response · Authors · 2023-11-17
>
> ### We thank Reviewer cs97 for their efforts in offering constructive comments and questions.
>
> 1. > **Reviewer**: The paper is in general not well written, especially in section 4. For example, in section 4.2, what are parameters of $\hat{p}_i^V$? And in section 4.3, are those $m_i^V$ are automatically generated or pre-selected and using LLM to get its likelihood?
>
> **Response**:
> We have gone through section 4 and made some updates to the notation, this has been re-uploaded and marked in red. To clarify for the reviewer, the parameters of $\hat{p}_i^V$ are the prompt defined by the environment ($p_i^V$) and the agent identifier $c_i^V$. This will output a new prompt, based on the original, that is updated to include the agent identifier. We have updated the paper such that the agent identifier now is a parameter of $p_i^V$ directly, removing the need for $\hat{p}_i^V$. This does not impact anything else.
>
> $m_i^V$ are pre-selected by us (i.e. we picked the discrete set of messages for each environment). This can be seen as a hyper-parameter for the environment, as different sets of available messages will impact the performance. This is an obvious limitation of the work, and an obvious extension is to allow the LLM to generate their own messages. However in practice we found the Flan models incapable of generating messages that were useful. We leave this to future work / stronger LLM models. Therefore, the $m_{i,t}^V$ picked by the agent at each time-step $t$ are picked in the same way as the actions are selected.
>
> 2. > **Reviewer**: Since it's a paper without any theoretical justification, more environments results might be more convincing.
>
> **Response**:
> Whilst we believe that our current set of environments are a strong indication of the benefits of our algorithm, and that they provide a wide coverage for justification, we will add some additional experiments to the camera-ready version of the paper. Please check the general comment we have left with a list of non-exhaustive additional experiments we will add.
>
>
> We hope we have managed to address the concerns of the reviewer. If so, we ask that the reviewer would consider improving their score, or to point out other additional issues that we can attempt to clarify.

---

> > ### Comment · Reviewer_cs97 · 2023-11-23
> >
> > Thanks for your feedback. I would like to keep my rating unchanged.

---

### Official Review · Reviewer_q4e1 · 2023-11-01

**Soundness:** 2 fair
**Presentation:** 2 fair
**Contribution:** 2 fair
**Rating:** 3
**Confidence:** 3

**Summary:**

The paper introduces the 'Functionally-Aligned Multi-Agents' (FAMA) framework towards better textual MARL agents, by tuning the LLM with MAPPO to act as a shared actor and centralized Critic, also integrating a communication module (using LLM) with discrete messages. The paper does experiments on the extended BabyAI-text environment and the traffic junction environment with Flan-T5-Base(small).

**Strengths:**

- The paper studies the important question of how to better leverage LLMs for cooperative MARL, and whether natural language communication between agents is useful for improving coordination and interpretability.

- The paper introduces a new framework to tune the pre-trained LLM with MAPPO to act as a shared actor and centralized Critic for better coordination functionality alignment and integrates a communication module with natural languages though in a discrete manner.

**Weaknesses:**

The experiment results are not very convincing.

- In Figure 3, there seems no significant difference between the proposed method and the baseline on the Junction Environment (individual agents!), which largely weakens the effectiveness of the proposed method from my perspective.

- If I'm not misunderstanding it,  only for experiments in Figure 3 Junction Environment the Small size version of the Flan-T5 is used. Then why is a larger size model (Base) used in Figure 4 with ablations?

- To answer the Q2 raised in the paper, the current results in Figure 4 are not enough to me, more analysis on the discrete message selected is needed.

- More training details on the baselines are needed.

- The environment experimented on seems simple and toysome with short horizons. It would strengthen the paper to have more experiments on harder envs.

The presentation is poor

- Figures need further improvements. Eg. In Figures 1 and 2, the text under the image is too small, in Figure 3, the text is too small and hard to tell whether a higher or lower metric is better.

- The notation used in sec 4.1 is not always consistent.

- There are many typos in the paper, especially the citations are in a weird format.

- How's the communication done when there are more than 2 agents (as shown in Figure 2 b)? How are the discrete messages selected in the first place and how does that affect the performance?

- Only using a simple textual agent identifier with a shared actor network may not work when the agents are not homogeneous

**Questions:**

Please address the concerns mentioned in the weaknesses.

---

> ### Author Response · Authors · 2023-11-17
>
> ### We thank Reviewer q4e1 for their efforts in offering constructive comments and questions.
>
> 1. > **Reviewer**: In Figure 3, proposed method vs. baseline.
>
> **Response**:
> We provide a table of the results so that it is easier to see the differences that may be masked by the scaling of the y-axis, and believe they demonstrate strong improvements towards to the optimum performance. We will include these details in the paper.
>
>
> |  | FAMA | Independent GLAM |
> | -------- | -------- | -------- |
> | Average Episode Return     | $4.58 \pm 0.36$     | $3.79 \pm 0.26$     |
> | -------- | -------- | -------- |
> | Average Episode Completion Rate     | $4.22 \pm 0.07$     | $4.05 \pm 0.009$     |
> | -------- | -------- | -------- |
> | Average Episode Crash Rate     | $0.68 \pm 0.06$     | $0.92 \pm 0.06$     |
>
>
> 2. > **Reviewer**: Why is a larger size model (Base) used in Figure 4 with ablations?
>
> **Response**:
>
> This is correct. We found that in testing the discrete message passing module Flan-T5 (Small) generally struggled to use the module correctly. Only when using Flan-T5 (Base) did we find the message passing module to be useful. For the camera-ready version, we will place the results for Flan-T5 into the appendix to demonstrate this.
>
> 3. > **Reviewer**: Current results in Figure 4 are not enough, more analysis on the discrete message selected is needed.
>
> **Response**:
> For the camera-ready version of the paper we will extend the discrete message selection analysis in the following ways;
>
> - We will add an ablation over the selection of the discrete messages (i.e. changing the set of possible discrete messages).
> - We will add performance results on a new, more difficult, variant of the junction environment.
>
>
> 4. > **Reviewer**: More training details on the baselines are needed.
>
> **Response**:
> We have added additional training details in Appendix B.3.
>
> 5. > **Reviewer**: The environment experimented on seems simple and toysome with short horizons.
>
> **Response**:
> We respectfully disagree with the reviewer and believe that the environments we run our experiments on are good benchmark environments. For example, the difficulty of the BabyAI environment is noted in the original BabyAI [BabyAI: A Platform to Study the Sample Efficiency of Grounded Language Learning - Chevalier-Boisvert et al. 2019] paper, showing that additional techniques such as imitation learning / curriculum learning are needed to perform well across the suite of tasks in BabyAI. We additionally can see that the multi-agent BabyAI-text environments can quickly become difficult (e.g. the PickUp task we use still shows relatively low success rates in Fig. 3A) without any forms of learning help (e.g. curriculum learning). In addition, we maintain a smaller view of the world for each agent (6x6 grid vs. 7x7 grid in original BabyAI) which increases the difficulty.
>
> The Junction environment is easier than the BabyAI environment, and is predominantely used to demonstrate the effectiveness of the communication module. We will provide more difficult variants of this environment for the camera-ready version of the paper, introducing more lanes, more active agents and longer time-horizons.
>
> 6. > **Reviewer**: Figures need further improvements.
>
> **Response**:
> We have updated this in the manuscript.
>
> 7. > **Reviewer**: The notation used in sec 4.1 is not always consistent.
>
> **Response**:
> We thank the reviewer for pointing this out. We have uploaded a new version of the paper where we have fixed the notation in Sec 4.1. The changes are marked in red.
>
> 8. > **Reviewer**: There are many typos in the paper, especially the citations are in a weird format.
>
> **Response**:
> We thank the reviewer for noting that there are typos. We have gone through the paper once again and made changes (all changes marked in red).
>
> 9. > **Reviewer**: How's the communication done when there are more than 2 agents (as shown in Figure 2 b)? How are the discrete messages selected in the first place and how does that affect the performance?
>
> **Response**:
>
> Our framework is designed to easily handle more than 2 agents. Specifically in terms of the communication module, it does not change for $N>2$ agents, apart from requiring a random ordering of message generation being applied (i.e. randomly select which agent sends the first message, then second etc...).
>
> The discrete messages are designed by us as the most logical forms of messaging in the environment in order to encourage coordinated behaviour. We recognise that this is a limitation, with the obvious next step being to remove discrete message selection and allowing the LLM to freely generate messages. However, we found in practice that the Flan-T5 models generally failed to generate coherent messages when not constrained to discrete options. The selection of the initial discrete messages can be thought of as like a hyper-parameter, and needs to be tuned to find the optimal set to maximise performance.

---

> > ### Author Response · Authors · 2023-11-17
> >
> > 10. > **Reviewer**: Simple textual agent identifier when the agents are not homogeneous
> >
> > **Response**:
> > We generally agree with the reviewer that this may be the case when agents are not homogenous. However, our goal was to validate our framework in some benchmark MARL setups, which in practice are typically limited to the homogenous setting. For example, all of the strictly cooperative MARL environments in PettingZoo have homogenous agents. We could extend the framework to cooperative team environments, which involve a competitive element which may remove homogeneity, however this is out of the scope of this work.
> >
> > We hope we have managed to address the concerns of the reviewer. If so, we ask that the reviewer would consider improving their score, or to point out other additional issues that we can attempt to clarify.

---

> > > ### Comment · Reviewer_q4e1 · 2023-11-23
> > >
> > > Thank you for your responses. However, the current experiment results are still not very convincing to me. I'll maintain my current score.

---

### Official Review · Reviewer_Mrhw · 2023-11-01

**Soundness:** 4 excellent
**Presentation:** 4 excellent
**Contribution:** 2 fair
**Rating:** 5
**Confidence:** 3

**Summary:**

This paper proposed FAMA which utilizes LLM-based agents in multi-agent settings, to solve the problems of sample inefficiency in online MARL training, policy generalization, and human interpretablity. Evaluations were done in two multi-agent textual environments.

**Strengths:**

It seems to be a novelty to consider LLM-based agents in MARL tasks, although the idea is straightforward in the context of LLM agent research.

**Weaknesses:**

From my viewpoint, there's a lack of deeper insights or discussion about why LLM-based agents work better in MAS tasks. See the Questions part.

**Questions:**

1. Although using LLM agents in MAS is a straightforward idea, I am still wondering: do we really need multiple LLMs to solve the problems? (Especially considering that the game settings in the paper are not fully decentralized.) Could the agents in the MAS task just send their partial observations to a single central LLM, which will make all the decisions? In my opinion, the aim of multiple LLM agents is more about exploring the potential of LLM, for example, when facing a complicated task, using an explicit planner agent and an explicit executor agent is better than throwing all the problems to a single LLM agent. But in this paper, LLM agents are just the agents in MAS tasks.
2. Prompt function piV, is it a fixed one or a trainable one?
3. Is there any idea about why policies output by finetuned LLMs perform better than the ones by traditional RL algorithms? After all, the experimental tasks are not complicated, and the RL algorithms should be specially designed for this.

---

> ### Author Response · Authors · 2023-11-17
>
> ### We thank Reviewer Mrhw for their efforts in offering constructive comments and questions.
>
> 1. > **Reviewer**: Do we really need multiple LLMs to solve the problems? (Especially considering that the game settings in the paper are not fully decentralised.)
>
> **Response**:
>
> We generally agree with the reviewer on this point. Even if the problem is decentralised, we should ideally be able to utilise a single set of LLM base weights for a multi-agent task (even just to reduce the computational practicalities of training an individual LLM per agent). One of the useful details of centralised multi-agent systems is that there is a lot of relevant shared information between agents, including basic capabilities that shouldn't need to be individually learned for each agent. However, in many scenarios it is important to maintain some form of individuality (e.g. a BabyAI task where agents have different roles towards the main goal) amongst agents.
>
> Our framework seems to be in line with what the reviewer is suggesting. Notably, our game setting is fully decentralised at evaluation time. The multiple agents all call to a *single* central LLM by passing their respective partial observations to it with an additional term to identify the agent (Sec. 4.1). During evaluation, this can be viewed as each agent having their own LLM, just with the same weights as others. Centralisation only occurs during training.
>
> 2. > **Reviewer**: Prompt function $p_i^V$, is it a fixed one or a trainable one?
>
> **Response**:
>
> In the current iteration of the paper, $p_i^V$ is fixed and designed by ourselves dependent on the task. We do not see a reason why it could not be trainable, however we do not necessarily believe a trainable prompt would be particularly useful for this task, nor is it within the scope of this paper. In the future, we believe this could be a worthwhile avenue of research.
>
>
> 3. > **Reviewer**: Is there any idea about why policies output by finetuned LLMs perform better than the ones by traditional RL algorithms?
>
> **Response**:
>
> Let us first consider our BabyAI problem setting, which focuses on two areas where we believe LLM policies can usefully replace RL policies:
>
> 1) Generalising across goals - In our BabyAI problem setting, a new goal object is drawn every episode (for example, in the GoTo task, the object that the agents need to go to changes each episode). LLMs are strong at generalising knowledge.
> 2) Sample efficiency - The BabyAI tasks are fairly sparse in terms of reward (e.g. only reward if successful at the end of an episode. Note, we do use some reward shaping to improve the reward density, however this is applied across all algorithms).
>
> As a combination, these two details make the training for our symbolic RL baselines more difficult. An LLM policy is well-designed to tackle both problems. Firstly, it can generalise across new objects far more easily than a symbolic policy, as it inherently already understands the connections between objects within its base knowledge.
> Secondly, the base knowledge also accounts for the LLM policy being sample efficient unlike the symbolic methods. An obvious impact of this is shown in Fig. 3 - the base knowledge of the LLM approaches can suceed in the experiments without any extra training (look at frame 0 of the results), whereas the symbolic approaches generally fail completely initially. This base knowledge can weaken the effect of the sparse reward, making the tasks solvable with <200K frames of training. This is not possible within this many frames for the symbolic approaches.
>
> We hope we have managed to address the concerns of the reviewer. If so, we ask that the reviewer would consider improving their score, or to point out other additional issues that we can attempt to clarify.

---

### Author Response · Authors · 2023-11-17

### General Comment
We thank all of the reviewers for taking the time to engage with our work. We believe the comments, and criticisms, are all useful in improving the quality of our work.

We have uploaded a new manuscript, with changes highlighted in red. The reviewers noted typos / notation inconsistencies, especially in Section 4, which we have hopefully addressed for the reviewers. In addition, the figures have been made clearer and a new appendix section (B.3) has been added to provide more baseline training details.

In addition, whilst we believe that our experimental results provide a wide coverage and show the effectiveness of our framework, we will add 3 new results to the camera-ready version of the paper:

1) A more difficult junction environment involving more lanes and more cars (agents).
2) An additional communication ablation over the discrete message available to the agents.
3) We will add the IC3Net baseline (Learning When To Communicate at Scale in Multiagent Cooperative and Competitive Tasks - Singh et al. ICLR 2019) to the communication experiments.

---

### Meta-Review · Area_Chair_bfBz · 2023-12-06

**Metareview:**

The paper proposes the Functionally-Aligned Multi-Agents (FAMA) framework to improve multi-agent coordination. FAMA employs by using Large Language Models (LLMs) for action selection and inter-agent communication, extending traditional games to text-based environments. FAMA is evaluated in two multi-agent textual environments: BabyAI-Text and traffic junction environment, showing improved performance over baselines.

Overall, although the use of LLMs as policy in multi-agent systems is relatively new, there are a few major weaknesses. First, the technique for alignment strategy does not differ significantly from past CTDE methods, making the contribution limited. Second, the experimental environments are too simple to support the necessity and contribution of the proposed method. For example,  current experimental settings do not motivate the necessity of decentralized agents. Third, no evidence supports the benefit of natural language communication over non-linguistic communication.

**Justification For Why Not Higher Score:**

The major weaknesses are not addressed, making the proposed method inconclusive.

**Justification For Why Not Lower Score:**

N/A

---

### Decision · Program_Chairs · 2024-01-16

Reject